# The Mechanism of Effort Intolerance in Patients with Peripheral Arterial Disease: A Combined Stress Echocardiography and Cardiopulmonary Exercise Test

**DOI:** 10.3390/jcm12185817

**Published:** 2023-09-07

**Authors:** Eihab Ghantous, Aviel Shetrit, Yonatan Erez, Natalie Noam, Ryan S. Zamanzadeh, David Zahler, Yoav Granot, Erez Levi, Michal Laufer Perl, Shmuel Banai, Yan Topilsky, Ofer Havakuk

**Affiliations:** 1Cardiology Division, Tel Aviv Sourasky Medical Center, Tel Aviv University, Tel Aviv 6997801, Israel; ehab.1988@yahoo.com (E.G.); aviel.shetrit@gmail.com (A.S.); yonierez@gmail.com (Y.E.); ryanzamanzadeh@gmail.com (R.S.Z.); david.zahler@gmail.com (D.Z.); yoavgran@gmail.com (Y.G.); erezool10@gmail.com (E.L.); michalpela@gmail.com (M.L.P.); shmuelb@tlvmc.gov.il (S.B.); topilskyyan@gmail.com (Y.T.); 2Vascular Surgery Department, Tel Aviv Sourasky Medical Center, Tel Aviv University, Tel Aviv 6997801, Israel; natalie_noam@hotmail.com

**Keywords:** peripheral arterial disease, stress echocardiography, cardiopulmonary exercise test, effort intolerance

## Abstract

Aim: We used a combined stress echocardiography and cardiopulmonary exercise test (CPET) to explore effort intolerance in peripheral arterial disease (PAD) patients. Methods: Twenty-three patients who had both PAD and coronary artery disease (CAD) were compared with twenty-four sex- and age-matched CAD patients and fifteen normal controls using a symptom-limited ramp bicycle CPET on a tilting dedicated ergometer. Echocardiographic images were obtained concurrently with gas exchange measurements along predefined stages of exercise. Oxygen extraction was calculated using the Fick equation at each activity level. Results: Along the stages of exercise (unloaded; anaerobic threshold; peak), in PAD + CAD patients compared with CAD or controls, diastolic function worsened (*p* = 0.051 and *p* = 0.013, respectively), and oxygen consumption (*p* < 0.001 and *p* < 0.001, respectively) and oxygen pulse (*p* = 0.0024 and *p* = 0.0027, respectively) were reduced. Notably, oxygen pulse was blunted due to an insufficient increase in both stroke volume (*p* = 0.025 and *p* = 0.028, respectively) and peripheral oxygen extraction (*p* = 0.031 and *p* = 0.038, respectively). Chronotropic incompetence was more prevalent in PAD patients and persisted after correction for beta-blocker use (62% vs. 42% and 11%, respectively). Conclusions: In PAD patients, exercise limitation is associated with diastolic dysfunction, chronotropic incompetence and peripheral factors.

## 1. Introduction

Peripheral arterial disease (PAD) affects 6.5% of people aged 50–55 years in the Western world, and its prevalence reaches 13% in those aged 70–75 years [1,2]. PAD is associated with a significant risk for cardiovascular outcomes, but is often characterized by effort intolerance, usually associated with lower extremity pain [1,2]. Studies exploring the mechanism behind effort intolerance in PAD are limited and, therefore, it is difficult to delineate its drivers and to suggest targeted therapies. In this study, we investigated effort intolerance in symptomatic PAD with the use of a combined stress echocardiography (SE) and cardiopulmonary exercise test (CPET). Due to the high prevalence of coronary artery disease (CAD) in PAD patients [3,4] and the potential effect of CAD on exercise tolerance [5,6], we used both CAD patients and normal controls as references.

## 2. Methods

### 2.1. Study Cohort

Consecutive PAD patients who also had CAD (i.e., those who underwent a percutaneous coronary intervention or had a history of an acute myocardial infarction) and experienced from lower extremity pain underwent combined SE with CPET as part of an investigation for effort intolerance at our center. Diagnosis of PAD was based on the following: extra-cranial carotid artery stenosis > 50%, abdominal aortic aneurysm, ankle–brachial index (ABI) < 0.9 or the need for peripheral arterial revascularization [1]. All patients were previously followed by a vascular surgeon and had complaints of exercise-induced leg pain. Patients were excluded from the study if they could not complete the exercise test (i.e., respiratory exchange ratio (RER) < 1, n = 7), had an abnormal echocardiographic exam (i.e., >moderate valvular heart disease (n = 2), a left ventricular ejection fraction (LVEF) < 40% (n = 4)) or had severe pulmonary disease (n = 1) (“PAD + CAD group”, n = 23).

Two groups of consecutive age- and sex-matched patients who were referred for effort dyspnea assessment during the study period were used as controls: (I) patients with established CAD (of whom 4 were excluded due to an inadequate exercise test and 4 were excluded due to abnormal rest echocardiography) (“CAD group”, n = 24), and (II) 15 normal controls. The 3 groups were matched according to age and sex, and the PAD + CAD group was matched with the CAD group according to major comorbidities, as described below. All study participants were ambulatory and stable. The trial was approved by our local ethics committee and all patients signed an informed consent.

### 2.2. Study Protocol

The study protocol included a combined SE with CPET and was previously described [7].

### 2.3. Cardiopulmonary Exercise Test

A symptom-limited graded ramp bicycle exercise test was performed in the semi-supine position on a tilting dedicated microprocessor-controlled eddy current brake SE cycle ergometer (Ergoselect 1000 L, CareFusion, San Diego, CA, USA). Expected maximal oxygen consumption (VO_2_ max) was the highest averaged 30 s VO_2_ during exercise and was estimated on the basis of patients’ age, height, weight and sex. Work rate increment necessary to reach patients’ estimated VO_2_ max in 8 to 12 min was then calculated. The protocol included 3 min of unloaded pedaling, a symptom-limited ramp graded exercise and 2 min of recovery. Breath-by-breath minute ventilation, carbon dioxide production (VCO_2_) and oxygen consumption (VO_2_) were measured using a Medical Graphics metabolic cart (ZAN, nSpire Health Inc., Oberthulba, Germany). The RER was defined as the ratio between VCO_2_ and VO_2_ obtained from ventilatory expired gas analysis [8]. Anaerobic threshold (AT) was determined manually using the modified V-slope method. Patients were encouraged to continue baseline medical therapy. In patients on beta-blocker therapy, chronotropic incompetence was determined when <62% of heart rate (HR) reserve was used [9].

### 2.4. Stress Echocardiography

Rest-stage echocardiographic images were taken before cycling ((iE33, Philips Medical Systems, Bothell, WA). Then, echocardiographic images were obtained concurrently with breath-by-breath gas exchange measurements in a continuous manner along predefined stages of exercise: rest; unloaded cycling; AT; and peak exercise (i.e., VO_2_ max). Each cycle of imaging included left ventricular end-diastolic and end-systolic volumes, stroke volume (SV), peak E- and A-wave velocities, deceleration time and septal e′, and lasted 30 to 60 s. LV end-diastolic volume, end-systolic volume and LVEF were calculated according to the single-plane ellipsoid apical 4-chamber area-length method. Left atrial volume was calculated according to the biplane area-length method [10]. SV was calculated by multiplying the LV outflow tract area at rest by the LV outflow tract velocity–time integral measured with pulsed-wave Doppler during each activity level. All echocardiographic measurements were performed using manual tracing [10]. Echocardiographic data were then analyzed retrospectively at different exercise stages. Unloaded-stage images were taken during unloaded exercise. AT (and the heart rate (HR) at the AT) was validated manually and retrospectively using the gas exchange measurements and the modified V-slope method. On the basis of HR at the AT echocardiogram, data were analyzed from the images captured immediately after reaching the HR at the AT. Peak exercise images were defined as those captured immediately after reaching RER > 1.05 [8]. Arteriovenous oxygen extraction (A–VO_2_) was calculated using the Fick equation as follows: (VO_2_)/(echocardiography-calculated CO) at each activity level [11].

## 3. Statistical Analysis

Categorical variables were summarized as frequency and percentage. Continuous variables were described as median and IQR. In order to compare clinical, echocardiographic and metabolic parameters in patients with PAD + CAD vs. CAD or normal controls, the groups were first matched according to age, sex, chronic kidney disease, diabetes, hypertension, LVEF and extent of CAD. The McNemar test was used to compare categorical variables and the Wilcoxon test was conducted to compare continuous variables. To analyze the differences at the different stages of effort, we used the repeated-measures linear model analysis to define the within-group effect for each parameter over time, the between-group differences over time and the group by time interactions. All statistical tests were two-sided and *p* < 0.05 was considered as statistically significant. SPSS software was used for all statistical analyses (IBM SPSS statistic for Windows, version 27, IBM corp., Armonk, NY, USA, 2020).

## 4. Results

The study group comprised 23 PAD + CAD patients. The mean age was 74.5 ± 7.1 years, 82% men, 92% had a history of smoking, the mean LVEF was 55 ± 6.3% and the mean estimated glomerular filtration rate was 63.4 ± 8.1 mL/min. The baseline characteristics of the three groups are shown in Table 1. Importantly, the extent of CAD was similar in the PAD + CAD vs. the CAD group. Nevertheless, baseline diastolic-function (E/E′ ratio: 12.1 ± 2.7 vs. 10.2 ± 2.4 vs. 8.3 ± 3.1, *p* = 0.023 and *p* = 0.012) and pulmonary-function tests (% predicted forced expiratory volume in 1 s: 79.1 ± 16.8 vs. 94 ± 15 vs. 96.2 ± 17.7, *p* = 0.03 and *p* = 0.035) were worse in the PAD + CAD group vs. the CAD or normal controls, respectively.

Along the stages of exercise (unloaded; anaerobic threshold; peak), comparing PAD + CAD patients vs. CAD or controls, cardiac output was reduced (*p* = 0.78 and *p* = 0.39; *p* = 0.10 and *p* = 0.002; *p* = 0.032 and *p* = 0.0011) due to a combination of blunted stroke volume increase (*p* = 0.7 and *p* = 0.61; *p* = 0.042 and *p* = 0.051; *p* = 0.036 and *p* = 0.039), impeded diastolic function (E/e′: *p* = 0.0014 and *p* = 0.007; *p* < 0.001 and *p* < 0.001; *p* = 0.0027 and *p* < 0.001) and diminished heart rate reserve (*p* = 0.28 and *p* = 0.19; *p* = 0.4 and *p* = 0.0013; *p* = 0.035 and *p* < 0.001), which persisted after correction for beta-blocker use (chronotropic incompetence was present in 62% vs. 42% and 11% of PAD + CAD vs. CAD patients or controls, *p* = 0.038 and *p* < 0.001, respectively) (full data in Table 2, Figure 1 and Figure 2). Furthermore, oxygen consumption was reduced (*p* = 0.26 and *p* = 0.19; *p* = 0.029 and *p* = 0.0012; *p* < 0.001 and *p* < 0.001) and oxygen pulse was blunted (*p* = 0.91 and *p* = 0.63; *p* = 0.064 and *p* = 0.085; *p* = 0.0021 and *p* < 0.001) not only due to an insufficient increase in stroke volume, but also because of a reduced oxygen extraction (*p* = 0.22 and *p* = 0.29; *p* = 0.041 and *p* = 0.049; *p* = 0.0015 and *p* = 0.002) (Table 2, Figure 1). Reduced ventilatory efficiency (peak VE/VCO_2_ 36.8 ± 4.9 vs. 31.5 ± 4 vs. 30.5 ± 5, respectively, *p* = 0.0031 and *p* = 0.0018) and reduced mechanical efficiency (slope of oxygen consumption/work rate 8.96 ± 0.42 vs. 10.37 ± 0.41 vs. 10.8 ± 0.49, respectively, *p* < 0.001 and *p* < 0.001, Figure 3) were found.

## 5. Discussion

Using an elaborate protocol, we have demonstrated here that exercise limitation in symptomatic PAD is associated with a combination of central and peripheral cardiovascular factors including diastolic dysfunction, diminished heart rate reserve and limited muscle oxygen extraction.

PAD is characterized by effort intolerance, which is often associated with intermittent claudication and leg pain [12]. This intuitive association is strengthened by the beneficial effect of exercise training on walking distance and symptom improvement [13]. Nevertheless, despite this clinical improvement, exercise training has little effect on indices of peripheral arterial function, such as the ABI [14]. This apparent contradiction might be settled given our findings showing that not only peripheral, but also cardiac, factors are involved in the cardiovascular limitation in PAD.

### 5.1. Diastolic Dysfunction

PAD patients experience a high prevalence of cardiovascular risk factors, including diabetes mellitus and hypertension [1,2]. Furthermore, CAD often coincides with PAD [4,15]. All are in association with diastolic dysfunction. Nevertheless, baseline echocardiographic indices often do not fulfill diagnostic criteria for diastolic dysfunction in PAD patients [16]. Studies exploring effort intolerance etiologies have shown that diastolic dysfunction could be unmasked by exercise [17]. Our findings, suggesting “concealed” diastolic dysfunction in PAD which is exposed by exercise, are in concordance with these results and may shed light on the attenuation in stroke volume and CO increase shown here. Notably, despite an overall similar risk profile in our CAD group, a diastolic limitation was less common in this group, suggesting that PAD + CAD patients may experience a higher prevalence of microvascular myocardial disease, which may contribute to stress-induced diastolic dysfunction [18].

### 5.2. Peripheral Oxygen Extraction

Notwithstanding the important effect of CO on oxygen uptake along exercise, we have shown here that peripheral oxygen extraction was reduced in PAD + CAD patients and contributed to oxygen-uptake attenuation. A signal for this limitation was demonstrated as early as the unloaded pedaling phase. The importance and the contribution of peripheral oxygen extraction to total oxygen uptake increases as CO decreases [19] and requires normal peripheral microvasculature and mitochondrial function, which are both compromised in PAD [20]. Consequently, the combination of stroke volume attenuation shown here together with reduced peripheral oxygen extraction augments the limitation in total oxygen consumption and exercise capacity in PAD.

### 5.3. Mechanical Efficiency

The combination of exercise-induced diastolic dysfunction, blunting of cardiac output and reduced peripheral oxygen extraction also corresponds with the reduced ratio of oxygen uptake to workload in PAD + CAD patients found in our study. Usually, a constant ratio of ΔO_2_/ΔW (usually around 10.5–11) is found in both normal and cardiovascular patients undergoing CPET studies. Nevertheless, a reduced ratio of ΔO_2_/ΔW characterizes high-risk patients (e.g., severe heart failure, pulmonary vascular disease) and might be caused by the presence of microvascular dysfunction, a change in the ratio of muscle fiber type or mitochondrial dysfunction. Different studies have shown that microvascular dysfunction is prevalent in PAD [21,22]. Furthermore, both pathology-based studies [23,24] and magnetic resonance imaging [25,26] have demonstrated a change in muscle characteristics and in mitochondrial function in PAD. Our results highlight these findings and emphasize the complex relationship between central and peripheral factors in PAD.

### 5.4. Chronotropic Incompetence

We have demonstrated here that PAD + CAD patients experience a high prevalence of chronotropic incompetence (CI). CI was repeatedly shown to be associated with increased morbidity and mortality in cardiovascular patients [27] and, in this sense, our findings may shed light on another potential contributor to the poor outcomes shown in PAD. Additionally, CI is an important determinant of cardiac output and maximal oxygen consumption [28,29,30] and further explains effort intolerance in PAD. Animal and human studies have demonstrated that PAD is associated with an increased sympathetic tone [31,32]. This may partially explain the limited “sympathetic reserve” and, consequently, the CI which was observed here.

Our findings may contribute to potential therapeutic interventions in PAD. The main method of exercise training in PAD is a supervised exercise program (SEP), which has demonstrated symptomatic improvement, improved quality of life and increased walking distance in PAD [33]. It involves, however, at least 2 h of weekly exercise for a 12-week period and, therefore, requires adherence which is not often achieved [34]. A potential alternative, high-intensity interval training (HIIT), which was found to be beneficial in heart failure and in CAD, was not adequately evaluated in PAD [35]. It will be interesting to explore the potential cardiac vs. peripheral cardiovascular effects of these two methods, an approach which may shed further light on our findings and their interpretation. Furthermore, our results, showing a “concealed” diastolic dysfunction in PAD, may imply that novel therapeutics (e.g., type 2 sodium-glucose transporter inhibitors [36,37]) might prove to be beneficial in PAD. Similarly, the presence of microvascular dysfunction is highly likely in PAD and could be addressed by novel therapeutics [38,39]. Last, the CI demonstrated here was also found in HF and could be potentially mitigated by exercise training or implantable devices.

Our study has several limitations. First, we used a noninvasive approach for CO and filling pressure measurements, and we calculated the anaerobic threshold according to gas exchange alone. Nevertheless, our approach enabled us to examine “real-life” patients without the bias of including only those who agree to undergo an invasive procedure. Furthermore, the use of simultaneous echocardiography enabled us to measure dynamic changes in cardiac chambers’ size and in valvular function. Second, our study group comprised PAD + CAD patients and not PAD alone. We believe, however, that the very high prevalence of CAD in PAD patients cannot be ignored, and our approach makes our findings more translational to everyday clinical practice. Third, our age- and sex-matched normal control group comprised relatively elderly patients with a low-intermediate prevalence of cardiovascular risk factors. Nevertheless, comparing our high-risk study group with a group of young and healthy controls might have damaged the validity of our findings. Fourth, despite our efforts to appropriately match the PAD + CAD group with the two control groups, these patients still had a significantly higher prevalence of smoking, a factor which might have affected our results. Nevertheless, the magnitude of difference in oxygen consumption, oxygen pulse [40,41,42] and certainly in diastolic dysfunction and chronotropic incompetence cannot be attributed to smoking alone.

In conclusion, using a comprehensive cardiopulmonary and stress echo test, we have demonstrated that exercise intolerance in PAD is associated with diastolic dysfunction, chronotropic incompetence and reduced peripheral oxygen extraction. Further studies are needed to establish targeted therapeutic approaches which might mitigate these limitations.

## Figures and Tables

**Figure 1 jcm-12-05817-f001:**
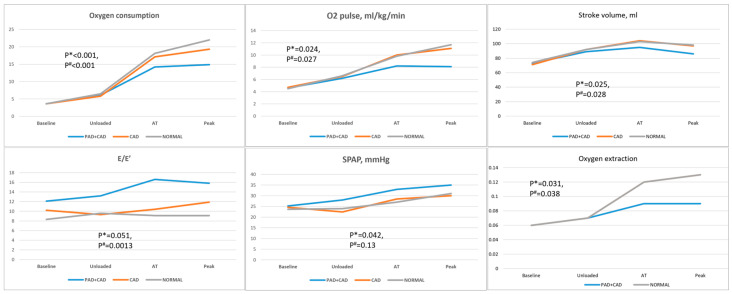
Along the stages of exercise, oxygen consumption was reduced in PAD patients due to a combination of mechanical and peripheral limitations. *p* *—PAD + CAD vs. CAD, *p* #—PAD + CAD vs. normal controls.

**Figure 2 jcm-12-05817-f002:**
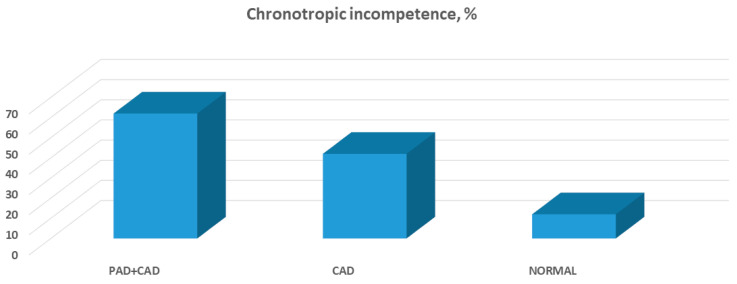
Compared with either CAD patients or normal controls, a higher prevalence of chronotropic incompetence was found among PAD + CAD patients.

**Figure 3 jcm-12-05817-f003:**
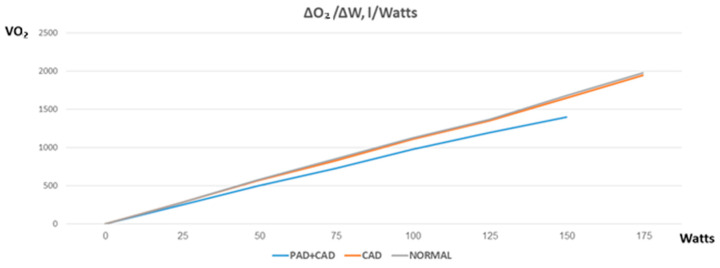
A decreased ratio of oxygen consumption to work rate was shown in PAD + CAD patients compared with either CAD patients or normal controls.

**Table 1 jcm-12-05817-t001:** Baseline (rest) characteristics.

	PAD + CAD n = 23	CAD n = 24	Normal Controls n = 15	*p* Value PAD + CAD vs. CAD	*p* Value PAD + CAD vs. Normal
Age, years	74.5 ± 7.1	72.9 ± 6.8	74 ± 7.7	0.21	0.86
Male, %	82	83	75	0.77	0.19
PAD involvement, %:					
CAS	26	NA	NA	NA	NA
AAA	13				
Ilio-femoral Rev.	48				
Low ABI	74				
Extent of CAD, %:			NA	0.11	NA
1-vessel disease	26	25			
2-vessel disease	61	69			
LM or 3-vessel disease	13	6			
SBP, mmHg	141 ± 8	138 ± 8	133 ± 11	0.26	0.03
Heart rate, bpm	71.1 ± 4.3	72.5 ± 5	78.7 ± 7.2	0.82	0.10
BMI, kg/m^2^	27.8 ± 5.2	27.3 ± 5.1	28.8	0.47	0.31
Current smoker, %	31	14	17	0.01	0.042
Past smoker, %	61	17	19	<0.001	<0.001
Hypertension, %	71	74	65	0.56	0.33
Diabetes mellitus, %	31	28	22	0.33	0.06
Hyperlipidemia, %	85	87	47	0.78	0.012
Atrial fibrillation, %	21	14	15	0.11	0.085
Creatinine, mg/dL	1.23 ± 0.46	1.07 ± 0.27	1.13 ± 0.39	0.29	0.2
Hemoglobin, g/dL	15.1 ± 2.1	14.7 ± 1.8	14.1 ± 2.7	0.29	0.31
Beta-blockers, %	76	68	26	0.22	0.0014
ACEI, %	60	65	68	0.2	0.26
Antiplatelet, %	88	92	12	0.6	<0.001
Statins, %	92	97	40	0.73	<0.001
EDV index, mL/m^2^	68 ± 9	67.1 ± 10.6	67.5 ± 11.5	0.54	0.71
ESV index, mL/m^2^	26 ± 8.4	25.2 ± 8.7	24.9 ± 8.9	0.81	0.63
Stroke volume, mL	73.3 ± 8.1	71.5 ± 8.5	74.3 ± 10.7	0.36	0.74
Cardiac output, L/min	5.4 ± 0.9	5.2 ± 0.85	5.4 ± 1.05	0.29	0.9
LVEF, %	55.1 ± 6.3	58.3 ± 4.5	62.1 ± 4.1	0.17	0.041
LVMI, g/m^2^	97.8 ± 26.1	94.9 ± 24.7	94.3 ± 27	0.43	0.71
LAVI, mL/m^2^	24.3 ± 6.8	21.2 ± 5.9	20.6 ± 5.1	0.15	0.11
E wave, m/s	0.64 ± 0.13	0.53 ± 0.1	0.57 ± 0.16	0.08	0.46
A wave, m/s	0.66 ± 0.06	0.61 ± 0.13	0.55 ± 0.1	0.19	0.04
E wave DT, m/s	204 ± 50	199 ± 47	201 ± 58	0.58	0.62
Averaged E′ cm/s	7.5 ± 2.3	9.0 ± 2.1	9.7 ± 3.1	0.17	0.07
E/E′ ratio	12.1 ± 2.7	10.2 ± 2.4	8.3 ± 3.1	0.18	0.013
S’, cm/s	8.2 ± 1.8	9.4 ± 1.9	9.4 ± 2	0.11	0.24
SPAP, mmHg	25.2 ± 4.2	24.6 ± 4.4	23.7 ± 4.9	0.51	0.31
FVC, L	3.2 ± 1.4	3.4 ± 1.1	3.4 ± 1	0.39	0.75
FVC, % predicted	103 ± 16.0	92.1 ± 14.5	94.5 ± 17	0.09	0.32
FEV_1_, L/s	2.6 ± 0.9	3.1 ± 1.1	3.1 ± 1.1	0.12	0.15
FEV_1_, % predicted	79.1 ± 16.8	94 ± 15	96.2 ± 17.7	0.05	0.03
FEV_1_/FVC	77 ± 8.1	92.1 ± 6.1	92.2 ± 6.3	0.01	0.01
VO_2_/kg/min	3.6 ± 0.8	3.6 ± 0.8	3.6 ± 0.85	1	0.96
RER	0.78 ± 0.07	0.79 ± 0.07	0.78 ± 0.08	0.91	0.93
AV difference, L/L	0.06 ± 0.02	0.06 ± 0.02	0.06 ± 0.02	1	1

Unless otherwise specified, numbers represent mean ± standard deviation. PAD—peripheral arterial disease, CAD—coronary artery disease, NA—not applicable, CAS—carotid artery stenosis, AAA—abdominal aortic aneurysm, Ilio-femoral Rev—history of ilio-femoral revascularization; ABI—ankle–brachial index, LM—left main, SBP—systolic blood pressure, BMI—body mass index, ACEI—angiotensin-converting enzyme inhibitors, EDV—end diastolic volume, ESV—end systolic volume, LVEF—left ventricular ejection fraction, LVMI—left ventricular mass index, LAVI—left atria volume index, E—mitral early flow wave, A—mitral late flow wave, DT—deceleration time, E′—mean mitral annulus tissue velocity, S′—lateral tricuspid valve tissue velocity, SPAP—systolic pulmonary artery pressure, FVC—functional vital capacity, FEV_1_—forced expiratory volume in 1 s, VO_2_—oxygen consumption, RER—respiratory exchange ratio, AV difference—arteriovenous oxygen extraction.

**Table 2 jcm-12-05817-t002:** Hemodynamic and metabolic indices along the stages of exercise.

	Baseline	Unloaded	AT	Peak	*p* Value within Group	*p* Value PAD + CAD vs. CAD	*p* Value PAD + CAD vs. Normal
EDV index, mL/m^2^ PAD + CAD CAD Normal	68 ± 9.4 67.1 ± 10.6 67.5 ± 11.5	76 ± 9.5 80.2 ± 10 80.7 ± 11.7	76.1 ± 9.5 86.6 ± 9.8 88. ± 12	75.3 ± 9.8 86.2 ± 10.2 85.9 ± 11.6	0.036 <0.001 <0.001	0.015	0.029
ESV index, mL/m^2^ PAD + CAD CAD Normal	26.3 ± 8.4 25.2 ± 8.7 24.9 ± 8.9	26 ± 9 24.5 ± 8.1 21.3 ± 9.6	24.3 ± 8.6 24.2 ± 9.5 23 ± 9.2	23.8 ± 8.1 21.1 ± 8.8 20.5 ± 9.7	0.36 0.17 0.27	0.44	0.39
Stroke volume, mL PAD + CAD CAD Normal	73.3 ± 8.1 71.5 ± 8.5 74.3 ± 10.7	89 ± 11 92 ± 11 92 ± 13	95 ± 10 104 ± 9 103 ± 11	86 ± 10 97 ± 10 98 ± 11	0.0014 <0.001 <0.001	0.025	0.028
HR, bpm PAD + CAD CAD Normal	71.1 ± 4.3 72.5 ± 5 78.7 ± 7.2	82 ± 12.6 86.3 ± 13.2 88.1 ± 14	97.3 ± 14.1 99.6 ± 13.3 116.7 ± 15	107.7 ± 17.2 119.8 ± 14.4 138.5 ± 18.9	<0.001 <0.001 <0.001	0.036	<0.001
CO, L/min PAD + CAD CAD Normal	5.4 ± 0.9 5.2 ± 0.85 5.4 ± 1.05	7.4 ± 2 7.5 ± 2.1 7.8 ± 3	9.3 ± 2.7 10.3 ± 2.4 11.9 ± 2.8	10.5 ± 2.9 12.7 ± 2.2 13.2 ± 3	<0.001 <0.001 <0.001	0.029	0.017
E/E′ ratio PAD + CAD CAD Normal	12.1 ± 2.7 10.2 ± 2.4 8.3 ± 3.1	13.2 ± 3.1 9.3 ± 2.4 9.6 ± 4	16.6 ± 4.3 10.4 ± 2.5 9.1 ± 3.8	15.8 ± 4 11.9 ± 2.9 9.1 ± 4.1	0.034 0.42 0.61	0.051	0.0013
S′, cm/s PAD + CAD CAD Normal	8.2 ± 1.8 9.4 ± 1.9 9.4 ± 2	8.1 ± 2 10.1 ± 2.1 9.7 ± 1.5	8.6 ± 1.6 9.6 ± 1.8 9.1 ± 1.8	8.8 ± 1.9 9.7 ± 2 9.6 ± 2.1	0.66 0.71 0.6	0.33	0.41
SPAP, mmHg PAD + CAD CAD Normal	25.2 ± 4.2 24.6 ± 4.4 23.7 ± 4.9	28.1 ± 4.9 24.2 ± 4.8 22.6 ± 5.2	32.9 ± 4.4 28.6 ± 3.8 27.1 ± 4.6	35 ± 5.1 30.2 ± 4 31.6 ± 6.3	<0.001 0.11 0.018	0.042	0.13
VO_2_, mL/kg/min PAD + CAD CAD Normal	3.6 ± 0.8 3.6 ± 0.8 3.6 ± 0.85	6.1 ± 1.4 5.8 ± 1.9 6.5 ± 2.5	14.2 ± 1.8 17.1 ± 2.9 18.1 ± 3.3	14.9 ± 2 19.3 ± 2.5 22 ± 3.6	<0.001 <0.001 <0.001	<0.001	<0.001
O_2_ pulse, mL/kg/min PAD + CAD CAD Normal	4.6 ± 1.3 4.7 ± 1.1 4.5 ± 1.3	6.2 ± 1.9 6.5 ± 2 6.6 ± 2.3	8.2 ± 1.8 10 ± 2.4 9.8 ± 2.3	8.1 ± 2.2 11.1 ± 2 11.7 ± 2.6	<0.001 <0.001 <0.001	0.0024	0.0027
AV difference, L/L PAD + CAD CAD Normal	0.06 ± 0.02 0.06 ± 0.02 0.06 ± 0.02	0.06 ± 0.03 0.07 ± 0.03 0.07 ± 0.03	0.09 ± 0.0 0.12 ± 0.0 0.12 ± 0.03	0.09 ± 0.0 0.13 ± 0.0 0.13 ± 0.03	0.043 <0.001 <0.001	0.031	0.038

Numbers represent mean ± standard deviation. AT—anaerobic threshold, EDV—end diastolic volume, PAD—peripheral arterial disease, CAD—coronary artery disease, ESV—end systolic volume, HR—heart rate, CO—cardiac output, SPAP—systolic pulmonary artery pressure, VO_2_—oxygen consumption, AV difference—arteriovenous oxygen extraction.

## Data Availability

The data underlying this article will be shared on reasonable request to the corresponding author.

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
