# Peer review of "The Mechanism of Effort Intolerance in Patients with Peripheral Arterial Disease: A Combined Stress Echocardiography and Cardiopulmonary Exercise Test"

_jcm, 2023, doi:10.3390/jcm12185817_

Round 1

Reviewer 1 Report

I think this is a great paper and congratulate the authors on the result. I only have a few (minor) comments to report:

- In the methods (line 54), I would remove "symptomatic" from "symptomatic lower extremity pain" (pain is symptomatic by definition)

- I think there is some confusion in the definition of PAD. In the manuscript (particularly in the discussion) when we talk about PAD we are essentially referring to lower extremity PAD, and all patients with CAD+PAD had lower extremity pain on exertion in the study. However, the definition of PAD in the paper did not include only lower extremity PAD, and only 74% of patients with CAD+PAD had ABI <0.9 (Table 1). How many had an ABI <0.9 and/or had lower limb revascularization? If it is not the totality of patients with CAD+PAD, I would use "lower extremity PAD" instead of "PAD" in the manuscript when referring specifically to this condition. On the other hand, if it is the totality of patients, I would define PAD as ABI <0.9 and/or lower extremity revascularization, and add the data on CAS and AAA as comorbidities. 

- What does "ilio-femoral" mean in Table 1? To avoid confusion, in Table 1 (PAD involvement) I would only report percentages of CAS, AAA, and lower extremity PAD (i.e., ABI<0.9 and/or lower extremity revascularization).

Author Response

We would like to thank the Editors and the Reviewers for their time and effort spent in reviewing our manuscript and for allowing us to improve it with their suggestions.

Reviewer 1:

I think this is a great paper and congratulate the authors on the result. I only have a few (minor) comments to report

RESPONSE: We thank the Reviewer for their kind words.

- In the methods (line 54), I would remove "symptomatic" from "symptomatic lower extremity pain" (pain is symptomatic by definition)

RESPONSE: Done.

- I think there is some confusion in the definition of PAD. In the manuscript (particularly in the discussion) when we talk about PAD we are essentially referring to lower extremity PAD, and all patients with CAD+PAD had lower extremity pain on exertion in the study. However, the definition of PAD in the paper did not include only lower extremity PAD, and only 74% of patients with CAD+PAD had ABI <0.9 (Table 1). How many had an ABI <0.9 and/or had lower limb revascularization? If it is not the totality of patients with CAD+PAD, I would use "lower extremity PAD" instead of "PAD" in the manuscript when referring specifically to this condition. On the other hand, if it is the totality of patients, I would define PAD as ABI <0.9 and/or lower extremity revascularization, and add the data on CAS and AAA as comorbidities. 

RESPONSE: The Reviewer is correct and, as shown in Table 1, not all PAD patients had lower extremity PAD. Interestingly, our Results suggest that most of these patients probably have "under-diagnosed" lower extremity PAD. However, we have changed our Discussion accordingly.

- What does "ilio-femoral" mean in Table 1? To avoid confusion, in Table 1 (PAD involvement) I would only report percentages of CAS, AAA, and lower extremity PAD (i.e., ABI<0.9 and/or lower extremity revascularization).

RESPONSE: We thank the Reviewer for allowing us to clarify this point. "ilio-femoral" stands for history of ilio-femoral revascularization. This was corrected in the Table.

Reviewer 2 Report

The manuscript is interesting and has practical value.

Page 2 line 56 Why did the authors choose carotid stenosis more 50% as a PAD diagnosis criterion? For example, why not 40%?

Was the diameter of the aortic aneurysm considered for PAD diagnosis?

Author Response

We would like to thank the Editors and the Reviewers for their time and effort spent in reviewing our manuscript and for allowing us to improve it with their suggestions.

Reviewer 2:

The manuscript is interesting and has practical value.

RESPONSE: We thank the Reviewer for their kind words.

Page 2 line 56 Why did the authors choose carotid stenosis more 50% as a PAD diagnosis criterion? For example, why not 40%?

RESPONSE: We have used the cutoff value chosen by the ESC guideline - European Heart Journal (2018) 39, 763–821 – see p. 778.

Was the diameter of the aortic aneurysm considered for PAD diagnosis?

RESPONSE: No. Data regarding the presence of AAA were driven from patients' files and verified with their treating physician.
